# Zoledronic Acid Ameliorates the Bone Turnover Activity and Periprosthetic Bone Preservation in Cementless Total Hip Arthroplasty

**DOI:** 10.3390/ph15040420

**Published:** 2022-03-30

**Authors:** Allen Herng Shouh Hsu, Chun-Hsien Yen, Feng-Chih Kuo, Cheng-Ta Wu, Tsan-Wen Huang, Juei-Tang Cheng, Mel S. Lee

**Affiliations:** 1Department of Orthopedic Surgery, Kaohsiung Chang Gung Memorial Hospital, College of Medicine, Chang Gung University, Kaohsiung City 833, Taiwan; alking0061@gmail.com (A.H.S.H.); fongchikuo@adm.cgmh.org.tw (F.-C.K.); oliverwu429@adm.cgmh.org.tw (C.-T.W.); 2Ministry of Health and Welfare Cishan Hospital, Kaohsiung City 842, Taiwan; acow690@gmail.com; 3Department of Orthopedic Surgery, Chia-Yi Chang Gung Memorial Hospital, College of Medicine, Chang Gung University, Chiayi City 621, Taiwan; b8601081@adm.cgmh.org.tw; 4Department of Medical Research, Chi-Mei Hospital, Tainan City 710, Taiwan

**Keywords:** bisphosphonate, zoledronic acid, total hip arthroplasty, bone mineral density, bone turnover marker

## Abstract

The administration of zoledronic acid (ZA) to patients who received cementless total hip arthroplasty (THA) has been reported to reduce bone turnover markers (BTMs) and increase bone mineral density (BMD). The effects of two-dose ZA versus placebo on cementless THA patients were analyzed in this five-year extension study. Alkaline phosphatase (ALP), osteocalcin (OC), procollagen 1 intact N-terminal propeptide (P1NP), serum calcium, renal function, radiological findings, and functional outcomes were compared in 49 patients, and the periprosthetic BMD of seven Gruen zones were compared in 19 patients. All the patients had normal renal function and calcium levels at their final follow-up. The mean ALP level in the ZA group was significantly lower at the fifth year, mean OC levels were significantly lower at the second and fifth year, and mean P1NP levels were significantly lower from 6 weeks to 5 years as compared with the control group. Fifth-year BMD levels were not found to be different between the ZA and control groups. The BMD Change Ratios in the ZA group were significantly increased in Gruen zone 6 at 1, 2, and 5 years. Our study results suggest that short-term ZA treatment with a subsequent 4-year drug holiday may inhibit serum BTMs and provide periprosthetic bone preservation at five years without adverse events.

## 1. Introduction

Zoledronic acid (ZA), a third-generation bisphosphonate, has been shown to be more potent than its second-generation predecessor in multiple in vivo studies [1,2,3]. ZA avidly binds to bone and rapidly inhibits the HMB-CoA reductase pathway [4]. In a 3-year extension of the HORIZON (Health Outcomes and Reduced Incidence with Zoledronic Acid Once Yearly) pivotal fracture trial, patients who discontinued the drug therapy for 3 years had a minimal difference in bone density and bone turnover markers (BTMs) compared with those in continuous therapy [5]. These findings suggest that the residual effects of ZA after a drug holiday may be associated with its high affinity and potency.

Periprosthetic bone loss after total hip arthroplasty (THA) is a perturbing phenomenon associated with stress shielding and bone remodeling, leading to a higher risk of periprosthetic fracture, implant migration, and implant loosening [6,7,8,9]. Several studies have suggested that the use of bisphosphonate after THA can preserve periprosthetic bone mass and improve long-term outcomes [10,11,12,13,14]. However, the specific treatment duration of bisphosphonate in THA patients and whether the effect can last after drug discontinuation remain unknown. Adverse events such as osteonecrosis of the jaw (ONJ) and atypical bony fractures are also of concern regarding the long-term safety of bisphosphonate treatment.

In two randomized controlled studies using ZA treatment after THA, periprosthetic bone mineral density (BMD) was effectively preserved after 2 years [15,16]. Currently, many studies demonstrate the role of bisphosphonate in the preservation of periprosthetic bone after THA [17,18]. In our previous study, 60 cases of cementless THA were randomized to receive either two doses of ZA (administered one day post-operatively and one year post-operatively) or a placebo [15]. After the two-dose ZA treatment, no anti-osteoporotic medication was administered to both groups for 4 years. It is of interest to know whether the inhibition of periprosthetic bone loss and BTMs after the discontinuation of ZA in THA patients would be lasting as observed in the HORIZON pivotal fracture trial. Thus, this extension study at five years post-THA was conducted to reassess BTMs, periprosthetic BMD, serum calcium, renal function, radiological findings, and functional outcomes. We hypothesize that the short-term ZA treatment with a subsequent 4-year drug holiday may inhibit serum BTMs and provide periprosthetic bone preservation at five years without adverse events.

## 2. Results

### 2.1. Patient Deposition

The period of patient enrollment was between January 2010 and August 2011, as previously described. At 2 years, 27 patients in each group had completed follow-up examinations. At the end of 5 years, two patients in the ZA group and three patients in the control group dropped out due to medical illnesses or an unwillingness to continue participation. A remaining recruitment of 25 patients (15 females/10 males; mean age 60.6 ± 11.9 years; mean BMI 26 ± 4) in the ZA group and 24 patients (13 females/11 males; mean age 58.8 ± 13.3 years; mean BMI 25 ± 5) in the control group underwent analysis for BTMs, serum biomarkers, radiological findings, and functional outcomes. The fifth-year BMD was acquired in 9 patients within the ZA group and 10 patients within the control group, as shown in Figure 1. Refusal due to radiation concerns resulted in a lower number of cases receiving BMD analysis.

### 2.2. Functional Outcomes and Adverse Events

The patients in both groups had comparable good to excellent functional outcomes at the final follow-up as compared with one year post-THA. The Harris Hip Score improved from 86 ± 7.1 to 87 ± 9.7 in the ZA group and from 85 ± 7.3 to 89 ± 3.6 in the control group, with *p*-values of 0.112 and 0.095, respectively. The UCLA activity score improved from 5.7 ± 1.1 to 7.2 ± 0.8 in the ZA group and from 5.5 ± 1.2 to 7.2 ± 0.7 in the control group, with *p*-values of 0.18 and 0.124, respectively. No statistical difference was observed between the ZA group and the control group in regard to functional outcomes. ZA-related adverse effects included fever in three patients and hypocalcemia in one patient. All patients were treated in the outpatient setting without further clinical symptoms. No residual adverse effects were noted at the fifth-year follow-up.

### 2.3. Serum Biomarkers and Bone Turnover Markers (BTMs)

A decrease in the mean serum ALP level was observed in the ZA group with significance at 12 weeks (*p* = 0.004) as compared with the control group. Similar to the cohort of the previous study, the second-year ALP level was not found to be significantly lower; however, the mean ALP level at 5 years was observed to be suppressed with statistical significance (*p* = 0.002). The mean OC level was also found to be significantly higher at the baseline in the ZA group (*p* = 0.044) in the five-year cohort. A gradual decrease in the mean OC level throughout each time-point was observed in the ZA group with significance at 2 years (*p* = 0.034) and 5 years (*p* = 0.007) as compared with the control group. The mean P1NP level was also found to be significantly higher in the ZA group at baseline (*p* = 0.03). Following the first dose of ZA, the mean P1NP level was decreased with a significant difference at every time-point from 6 weeks to 5 years compared with the control group (Table 1). No statistical difference was observed between the two groups in renal function and serum calcium level (Table 2).

### 2.4. Periprosthetic Bone Mineral Density

Nine patients in the ZA group and ten patients in the control group received a bone mineral density (BMD) assessment in the current study; similar baseline (pre-THA) and BMD values were observed in both groups. The mean BMD values were significantly higher for the ZA group in Gruen zone 6 at 1 year and in zones 1, 6, 7 at 2 years. The mean BMD values of the ZA group in all Gruen zones at 5 years were higher than the BMD values of the control group but to no statistical significance (Table 3). The BMD Change Ratios were significantly higher in the ZA group in Gruen zones 5 and 6 at 1 year, Gruen zone 6 at 2 years, and Gruen zone 6 at 5 years (Figure 2). In the ZA group, the BMD Change Ratios at 5 years in each Gruen zone showed a consistently positive value of greater than one, suggesting an increase from baseline, but to no statistical significance compared with the control group (Table 4).

### 2.5. Radiologic Evaluation

The standard pelvis anteroposterior view and lateral hip radiography were analyzed at all time intervals. All the implants were well fixed with radiological evidence of prosthesis bony ingrowth. No progressive radiolucent lines, osteolysis, subsidence, or migration of prostheses were observed in both groups. No patients developed a periprosthetic fracture, infection, or other implant-related complications at the fifth year.

## 3. Discussion

The effects of ZA have been shown to promote bone ingrowth onto implant devices, enhancing implant fixation via osteoclast-mediated strain-adaptive periprosthetic bone resorption [19]. ZA infusion has a limited protective effect on immediate post-operative periprosthetic bone preservation, where the stability of the implant is primarily based on the implant design and cortical bone structure. ZA acting on trabecular bone begins to promote osteointegration and bony ongrowth weeks to months following the initial infusion and the ability to reduce implant migration thereinafter [20,21]. In the literature and clinical practice, anti-osteoporotic agents, including ZA, have been widely prescribed to osteoporotic patients undergoing joint replacement.

BMD value is an important factor in the fixation and stability of the implanted prosthesis [22]. Early stem migration, rotation, and subsidence have been detected in osteoporotic bone as compared with bones of normal BMD value in patients undergoing THA [23]. In our previous study, the short-term use of ZA increased the periprosthetic BMD values in three of the seven Gruen zones at 2 years [15]. In the cohort of this extension study, the ZA group retained higher mean BMD values at 5 years as compared with the control group in all seven Gruen zones but to no statistical significance. However, fifth-year BMD Change Ratios in Gruen zone 6 were observed to be persistently higher with statistical significance in the ZA group as compared with the control group at 1 year, 2 years, and 5 years after THA. In a meta-analysis by Zhao et al., the BMD ratio was noted to have significance in more than three zones as early as 6 months and at 1 year, which is in line with our preliminary study in 2017 [15,21]. In the same meta-analysis, BMD Change Ratios at 5 years were observed to be significant in zones 6 and 7, and our data also revealed significance in zone 6.

Parallel to the findings of BMD preservation, the levels of BTMs (ALP, OC, and P1NP) were shown to be persistently suppressed in the ZA group after the discontinuation of ZA for 4 years. Different from the cohort of the two-year study [15] in which the baseline OC and P1NP levels were statistically similar between groups, higher baseline values of OC and P1NP in the ZA group were observed in the cohort of this extension study (*p* = 0.044 and *p* = 0.030, respectively), suggesting that the reduced number of study subjects in the five-year study altered the cohort baseline values. A high bone turnover status with increased serum OC and P1NP is associated with low BMD in the osteoporotic population [24,25]. With patients in the ZA group having higher bone turnover status at baseline, ZA given one day after THA was able to significantly inhibit P1NP and ALP levels as early as 6 weeks. With the administration of a second dose of ZA one year after THA, further inhibition of OC and P1NP levels at 2 years and ALP, OC, and P1NP levels at 5 years was observed in this study. OC and P1NP are useful BTMs for monitoring antiresorptive therapy and fracture risk assessment [26,27,28].

To the best of our knowledge, the interesting findings of the lasting effect of ZA on periprosthetic BMD and bone turnover after a drug holiday of 4 years have not been discovered before. This lasting effect may be related to ZA’s high affinity to bone and its high potency pharmacological nature, as seen in the extension study of the HORIZON pivotal fracture trial [29]. Other bisphosphonates (risedronate, alendronate, and pamidronate) have shown good periprosthetic bone preservation in early follow-ups but not in the mid-term, as the increased periprosthetic BMD would eventually decrease after the discontinuation of these aforementioned medications. Different from previous enteral bisphosphonates, ZA has the advantage of disregarding oral bioavailability, being an intravenous drug of higher relative potency. Furthermore, the annual intravenous administration in the clinical setting had better patient compliance and study protocol implementation versus a weekly oral regimen of alendronate. Producing no significant increase in adverse events regarding atypical fractures, ZA has become a widely studied antiosteoporosis medication for orthopedic-related diseases [5,18,30,31,32]. A limited-course ZA infusion regimen after THA has been used in multiple prospective controlled trial studies with a similar design to show the promising long-term results [15,20,33]. In a study of limited-course ZA with a treatment course equating to one year or less, a bone preservation effect observed via BMD values could be maintained by bisphosphonate for up to five years post-operatively in Gruen zones 6 and 7 [21]. In associated literature reviews, a reduction in overall fracture rate by administrating a single dose of ZA infusion was also noted [34,35]. Although the long-term effects on the cessation of bisphosphonates in patients receiving THA still require further clinical validation, patients with severe osteoporosis could have other compromised skeletal structures. Thus, some authors have suggested the lifelong administration of bisphosphonates is warranted, especially in patients with poor perioperative bone stock and those receiving revision arthroplasty [8,36]. For patients with adequate bone stock, a short-term infusion of ZA may provide sufficient mid-term implant survivorship, minimizing the adverse effects of long-term bisphosphonate use.

Our results are in accordance with other studies on ZA drug holiday, as it demonstrates a prolonged effect after discontinuation [37]. The findings in our study show that a 4-year drug holiday of ZA has been observed to inhibit BTMs, preserve periprosthetic BMD, and exhibit no adverse reactions such as the rebounding phenomenon seen in RANKL antagonist withdrawal, ONJ, or atypical bony fractures [38,39] However, no difference was found in regard to functional outcomes and implant survival. All implants were well fixed without loosening, migration, or pedestal formation at the final follow-up radiography. No patients received revision surgery for periprosthetic joint infection or periprosthetic fracture. Despite the lack of clinical significance regarding radiographic and functional parameters, we found the periprosthetic bone preservation potential, seen via BTMs and BMD, of short-term ZA treatment lasting up to 5 years encouraging.

This study has some limitations. Firstly, the number of cases in the cohort was small, with a 49-patient cohort completing the five-year clinical follow-up and only 19 patients completing the fifth-year BMD study. Thirty patients did not agree to the final BMD study for reasons concerning excessive radiation exposure produced by dual-energy X-ray absorptiometry and for other personal reasons. However, these patients were available to provide for all other aspects of the study and had well-functioning hips. This resulted in a drastic cutback in the number of BMD values acquired and resulted in a reduction in the power of the study. Secondly, the randomization of open-label ZA or placebo was not blinded by a dummy for control subjects. Thirdly, although good patient compliance was achieved with no other anti-osteoporotic medication administered during the study period, calcium and vitamin D supplement intake was not strictly controlled. Finally, this study reports the cross-sectional results at the fifth year in a cohort receiving two doses of ZA and a subsequent drug holiday of 4 years. These results may not be translated into other dosing regimens or projected for long-term outcomes. Although statistically significant BMD data at five years were not found throughout all parameters acquired via BMD measurements, we were able to provide data on BTMs that suggest the lasting bone-turnover suppression effect of ZA at five years on a marked level.

In regard to the surgical outcome, good prosthesis fixation, low complication rates, and satisfactory hip function scores within our cohort could not provide direct evidence of the protective effect of ZA. This may be due to a relatively small cohort, but it is also a sign of good patient compliance and post-THA well-being, as well as a strict surgical technique performed on a well-tested implant design. As additional support for evidence of a low complication rate of the selected stem, a 5-year prospective radiostereometric analysis on a cementless double-tapered femoral stem has been shown to provide radiographic evidence of subsidence equal to or less than 2 mm in 4 of 51 hips for which none required revision surgery [40].

Despite our mid-term findings, the long-term efficacy of ZA infusion and its protective ability against periprosthetic fracture, stem subsidence, and risk of revision still require further validation. However, there are merits to this extension study in that the data of the prospective randomized clinical trial are unbiased and unique, and all surgeries were performed by an experienced surgeon. The study was stringently performed and audited by the institutional research board. With propitious results observed at the fifth year, a 10-year follow-up may be warranted to assess the long-term bone preservation effect of a two-dose ZA on cementless THA.

## 4. Materials and Methods

This five-year extension study of a prospective, randomized clinical trial on ZA treatment in patients undergoing cementless THA (ClinicalTrials.gov: NCT02838121, accessed on 19 July 2016) was conducted with Institutional Review Board approval (98-1150A3; 105-1296C1; 105-7004D). All patients within the study cohort received THA with Trilogy acetabular cup (Zimmer, Warsaw, IN, USA) and cementless VerSys Fiber Metal Taper femoral stem (Zimmer, Warsaw, IN, USA). Our original randomized controlled trial study included 30 patients in ZA group who received intravenous 5 mg ZA (Novartis Pharmaceuticals Corporation, Basel, Switzerland) at one day and at one year after THA and a control group of 30 patients who received placebo of intravenous saline infusion. A cohort of 54 patients completed the two-year follow-up study. This extension study comprised 49 patients from the original study completing a five-year follow-up without receiving further anti-osteoporotic medications. Data of the 49-patient cohort were reassessed from initial enrollment up to the fifth year. BTMs obtained include alkaline phosphatase (ALP) via spectrophotometric assay, osteocalcin (OC) via radioimmunoassay, and procollagen 1 intact N-terminal propeptide (P1NP) via enzyme-linked immunosorbent assay. Additionally, serum biomarkers, including serum creatinine and serum calcium, radiological examination (pelvis anteroposterior view and hip lateral view), and functional outcome were also determined (Harris Hip Score and University of California at Los Angeles (UCLA) activity score). Fifth-year periprosthetic BMD values of the seven Gruen zones were assessed for only 19 of the 49 patients via dual-energy X-ray absorptiometry (DEXA, Hologic Inc., Waltham, MA, USA).

The periprosthetic BMDs for all Gruen zones of the proximal femur were quantified as previously described [15]. Time-dependent changes in BMD with reference to pre-operative (baseline) BMD in each Gruen zone were defined as the BMD Change Ratios; a method of evaluating the change in BMD values from a time-point minus the baseline value has been used in many studies of similar design in the reviewed literature [15,16,17,18,41]. Radiological assessments include implant position, presence of prosthesis–bone interface radiolucent lines, and presence of implant subsidence, loosening, or migration [20,42]. Each post-operative radiograph and all patient functional scores were obtained and reviewed individually by two orthopedic surgeons at our institution.

Statistical analysis was performed using SPSS Statistics (version 19, IBM, Armonk, NY, USA). A generalized estimating equation was conducted at each time-point to test for differences in all parameters. Significance was set at *p*-value less than 0.05.

## 5. Conclusions

Our study demonstrates the effect of a short-term two-dose ZA regimen administered within one year of cementless THA. This regimen was able to affect bone metabolism by inhibiting OC and P1NP at 2 years and ALP, OC, and P1NP at 5 years. Furthermore, fifth-year BMD showed a consistent increase across all Gruen zones, with a statistically significant increase in BMD Change Ratios in Gruen zone 6 at 1 year, 2 years and 5 years. The short-term dosing of ZA followed by a drug holiday of 4 years has no systemic nor periprosthetic adverse events and shows a lasting effect on periprosthetic bone preservation.

## Figures and Tables

**Figure 1 pharmaceuticals-15-00420-f001:**
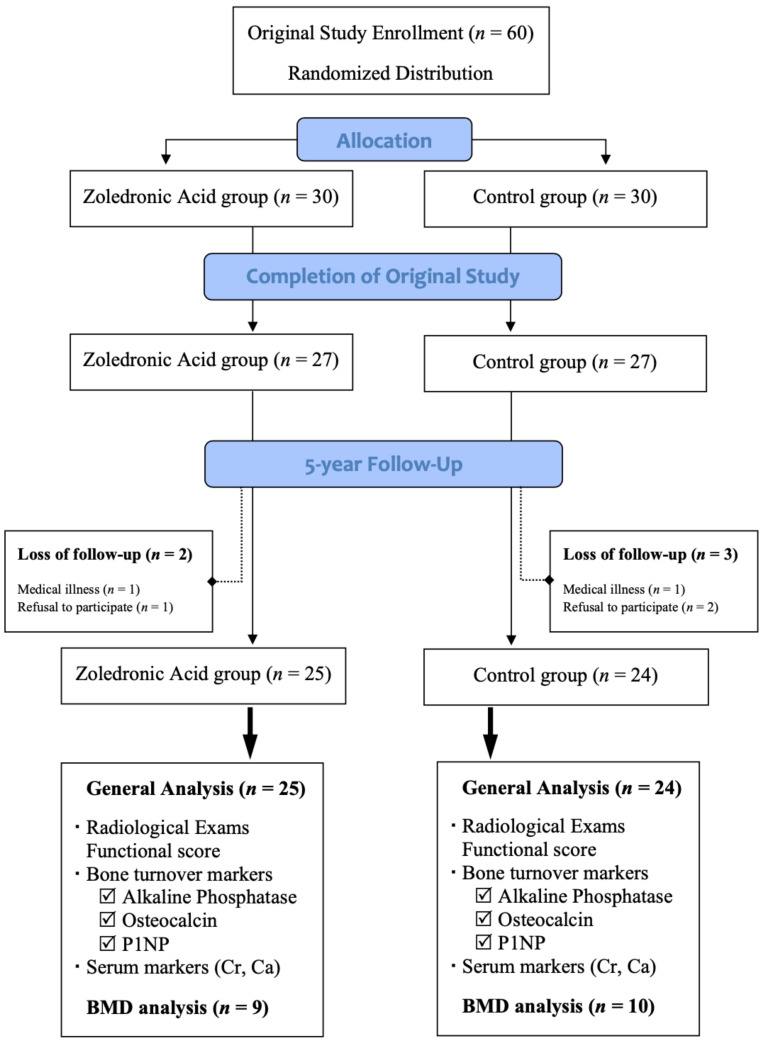
Patient deposition.

**Figure 2 pharmaceuticals-15-00420-f002:**
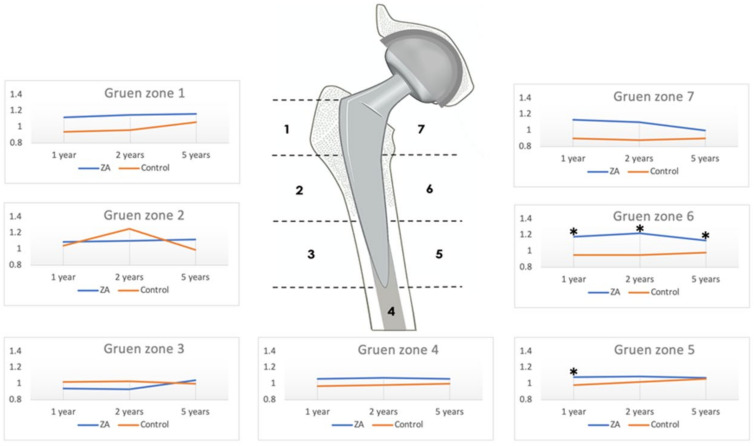
Bone mineral density Change Ratios for zoledronic acid (ZA) and control group in the seven Gruen zones. * *p* < 0.05.

**Table 1 pharmaceuticals-15-00420-t001:** Comparison of bone turnover markers in ZA (*n* = 25) and control (*n* = 24).

	Baseline	6 Weeks	12 Weeks	6 Months	1 Year	2 Years	5 Years
Alkaline phosphatase (µg/L)							
ZA	79.8 (17.9)	78.3 (19.4)	66.6 (11.4)	69.1 (27)	69.4 (24.7)	66.6 (17.5)	64.3 (11.3)
Control	76 (18.3)	87.9 (21.9)	81.9 (22.4)	74.8 (20.6)	74.6 (18.4)	74.5 (15.6)	76.7 (14.7)
*p*-value	0.091	0.062	0.004 *	0.095	0.088	0.060	0.002 *
Osteocalcin (µg/mL)							
ZA	21.9 (8.6)	18.3 (8.2)	17.3 (10.6)	14.8 (5)	16.9 (7.1)	14.5 (7)	12.5 (6.8)
Control	17.1 (7.4)	19.9 (12.2)	20.1 (10.8)	17.1 (5.4)	21.3 (9.8)	19.4 (8.5)	19.1 (8.6)
*p*-value	0.044 *	0.11	0.083	0.087	0.078	0.034 *	0.007 *
P1NP (procollagen 1 intact N-terminal) (ng/mL)							
ZA	55.2 (23.1)	55.5 (22)	37.5 (14.8)	30 (12.2)	35.2 (19.7)	27.7 (11.4)	24.3 (13.2)
Control	41.3 (20.2)	77.7 (39.3)	70.2 (37.5)	58.9 (25.1)	51.6 (34)	45.4 (19.6)	44.1 (20.4)
*p*-value	0.030 *	0.020 *	0.000 *	0.000 *	0.047 *	0.001 *	0.020 *

Data presented as mean (standard deviation), * *p* < 0.05.

**Table 2 pharmaceuticals-15-00420-t002:** Comparison of serum biomarkers between ZA (*n* = 25) and control (*n* = 24).

	Baseline	6 Weeks	12 Weeks	6 Months	1 Year	2 Years	5 Years
Creatinine (mg/dL)							
ZA	0.77 (0.21)	0.74 (0.24)	0.8 (0.28)	0.84 (0.61)	0.77 (0.25)	0.78 (0.25)	0.82 (0.27)
Control	0.82 (0.2)	0.79 (0.22)	0.78 (0.18)	0.78 (0.19)	0.81 (0.2)	0.81 (0.21)	0.82 (0.21)
GFR (mL/min)							
ZA	59.7 (1.7)	60.5 (4.8)	62 (11.1)	79.1 (27.7)	84.8 (28.6)	94.7 (22.6)	90.8 (25.8)
Control	59.8 (0.6)	59.8 (1.2)	63.6 (13.3)	73.7 (24.1)	81.4 (21.8)	87.7 (18.4)	85.9 (21)
Calcium (mg/dL)							
ZA	9.51 (0.5)	9.27 (0.53)	9.52 (0.45)	9.44 (0.46)	9.41 (0.48)	9.37 (0.37)	9.25 (0.4)
Control	9.6 (0.43)	9.16 (1.88)	9.53 (0.37)	9.43 (0.51)	9.48 (0.36)	9.28 (0.3)	9.38 (0.42)

Data presented as mean (standard deviation).

**Table 3 pharmaceuticals-15-00420-t003:** Mean BMD (g/cm^2^) for ZA (*n* = 9) and control (*n* = 10) by Gruen zones.

Gruen Zone	1	2	3	4	5	6	7
Baseline							
ZA	0.65 (0.12)	1.26 (0.22)	1.51 (0.38)	1.66 (0.25)	1.62 (0.22)	1.26 (0.19)	0.99 (0.17)
Control	0.63 (0.16)	1.28 (0.23)	1.57 (0.25)	1.69 (0.18)	1.65 (0.21)	1.30 (0.23)	0.95 (0.25)
*p*-value	0.767	0.806	0.708	0.784	0.761	0.703	0.713
1 year							
ZA	0.71 (0.08)	1.39 (0.15)	1.48 (0.35)	1.76 (0.23)	1.72 (0.17)	1.43 (0.15)	1.04 (0.20)
Control	0.57 (0.20)	1.30 (0.34)	1.57 (0.30)	1.62 (0.20)	1.58 (0.17)	1.19 (0.30)	0.80 (0.29)
*p*-value	0.072	0.437	0.552	0.199	0.096	0.047 *	0.054
2 years							
ZA	0.70 (0. 80)	1.37 (0.19)	1.43 (0.36)	1.75 (0.21)	1.72 (0.16)	1.46 (0.13)	0.99 (0.13)
Control	0.57 (0.14)	1.60 (1.06)	1.58 (0.23)	1.63 (0.20)	1.64 (0.21)	1.21 (0.28)	0.79 (0.23)
*p*-value	0.033 *	0.552	0.290	0.278	0.388	0.032 *	0.043 *
5 years							
ZA	0.72 (0.10)	1.41 (0.14)	1.61 (0.14)	1.76 (0.19)	1.72 (0.16)	1.38 (0.21)	0.93 (0.27)
Control	0.63 (0.16)	1.23 (0.28)	1.52 (0.20)	1.66 (0.21)	1.71 (0.26)	1.24 (0.26)	0.81 (0.21)
*p*-value	0.154	0.098	0.317	0.305	0.962	0.239	0.297

Data presented as mean (standard deviation), * *p* < 0.05.

**Table 4 pharmaceuticals-15-00420-t004:** Mean BMD Change Ratios by Gruen zone for ZA (*n* = 9) and control (*n* = 10).

Gruen Zone	1	2	3	4	5	6	7
1 year							
ZA	1.12 (0.28)	1.09 (0.11)	0.94 (0.21)	1.06 (0.13)	1.08 (0.11)	1.18 (0.18)	1.13 (0.31)
Control	0.94 (0.22)	1.04 (0.20)	1.02 (0.14)	0.97(0.08)	0.98 (0.07)	0.95 (0.23)	0.90 (0.32)
*p*-value	0.140	0.202	0.958	0.116	0.026 *	0.031 *	0.094
2 years							
ZA	1.15 (0.33)	1.10 (0.12)	0.93 (0.23)	1.07 (0.18)	1.09 (0.13)	1.22 (0.16)	1.10 (0.29)
Control	0.96 (0.14)	1.25 (0.64)	1.03 (0.09)	0.98 (0.08)	1.02 (0.06)	0.95 (0.18)	0.88 (0.25)
*p*-value	0.114	0.524	0.233	0.097	0.234	0.005 *	0.109
5 years							
ZA	1.16 (0.36)	1.12 (0.17)	1.04 (0.17)	1.06 (0.12)	1.07 (0.12)	1.13 (0.16)	1.00 (0.34)
Control	1.06 (0.27)	0.99 (0.19)	1.00 (0.14)	1.00 (0.09)	1.06 (0.10)	0.98 (0.11)	0.90 (0.19)
*p*-value	0.485	0.158	0.596	0.190	0.811	0.032 *	0.465

Data presented as mean (standard deviation), **p* < 0.05.

## Data Availability

Data is contained within the article.

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
