# Peer review of "Zoledronic Acid Ameliorates the Bone Turnover Activity and Periprosthetic Bone Preservation in Cementless Total Hip Arthroplasty"

_pharmaceuticals, 2022, doi:10.3390/ph15040420_

Round 1
Reviewer 1 Report
There is a lot of work on a similar topic. This work adds nothing new. This work is on a small number of patients (19 people). Other similar studies assess a larger number of patients.
Author Response
We thank the esteemed reviewer for your valued comments. We have updated the revised manuscript to include further supplementations and clarifications in hopes to refined and improve our study.

Reviewer 2 Report
Dear Authors,
I had the opportunity to revise your manuscript to know whether the inhibition of periprosthetic bone loss and BMTs after ZA discontinuation on THA patients would last after five years of follow-up.
In general, I found the manuscript interesting and well described, although some methodological aspects need improvement.
Line 27: Seem not correct to say that BMD levels were higher in the ZA group without a significant difference; I suggest saying that no differences were found in BDM levels between control and ZA groups.
Line 82-85:
- Harris Hip Score improved from 86 ± 7.1 to 87 ± 9.7 in ZA group and from 85 ± 7.3 to 89 ± 3.6 in control group.
- UCLA activity score improved from 5.7 ± 1.1 to 7.2 ± 0.8 in ZA group and from 5.5 ± 1.2 to 7.2 ± 0.7 in 84 control group.
It seems like the improvement is not relevant (CI collide) and that HHS keeps unchanged in both groups after one year of THA. I have doubts about the pre-post change on UCLA activity.
Please, indicate the p-value and the name of the test conducted in each comparison.
Tables:
Please, add the p-value of all the comparisons (in all the tables), even if they are not significant. The reason is that marginal differences or differences with a significance from 5% to 10% could be helpful to other researchers (i.e. The mean difference of table 3, after one year, in region one, is significant at 7%).
In table 2, remove the symbol (*) and meaning as is not used in the table data.
Line 148: Reshape this sentence: "In the cohort of this extension study, ZA group retained higher mean BMD values in all Gruen zones at 5 years statistical significance."
Line 216: This study, although well-conducted, does not have the power to detect superiority. Data are not showing the confidence intervals to make readers hypothesize this. Please, consider reformulating this sentence.
Line 269. A GEE was conducted at each time to test differences in all parameters.
Please, indicate if the regression model was univariate or multivariate. Whatever the case, specify the dependent and independent(s) variables(s).
Indicate the parameters used in the model in therm of link(identity, log, etc.), family (Gaussian, binomial, etc.) and type of correlation (unstructured, independent, exchangeable).
Does the model fit? Please indicate the model adjustment information.
If data were analyzed using univariate models, each table must show the model-adjustment information.
Thank you!
Author Response
Reviewer#2
I had the opportunity to revise your manuscript to know whether the inhibition of periprosthetic bone loss and BMTs after ZA discontinuation on THA patients would last after five years of follow-up.
In general, I found the manuscript interesting and well described, although some methodological aspects need improvement.
Line 27: Seem not correct to say that BMD levels were higher in the ZA group without a significant difference; I suggest saying that no differences were found in BDM levels between control and ZA groups.
Reply: We have made changes to the draft for rewording and clarification as advised. (revised Line 27)
Line 82-85:
Harris Hip Score improved from 86 ± 7.1 to 87 ± 9.7 in ZA group and from 85 ± 7.3 to 89 ± 3.6 in control group.
UCLA activity score improved from 5.7 ± 1.1 to 7.2 ± 0.8 in ZA group and from 5.5 ± 1.2 to 7.2 ± 0.7 in 84 control group.
It seems like the improvement is not relevant (CI collide) and that HHS keeps unchanged in both groups after one year of THA. I have doubts about the pre-post change on UCLA activity. Please, indicate the p-value and the name of the test conducted in each comparison.
Reply: Thank you for bringing up the issue of hip functional scores. We have supplemented the p-values for both UCLA activity score and Harris Hip score in the manuscript (revised line 86-90)
We would like to clarify that the hip score comparison was done for the first time one year after the hip replacement, with the assumption of complete recovery from the surgical treatment and a 12-month healing time for implant-bone ingrowth, as well as adequate return of daily hip function. The second time point for comparing hip scores was obtained 5 years after the hip replacement. The clinical significance of this time difference was to monitor the continued functionality of the hip implants in both the control and treatment groups.
Tables:
Please, add the p-value of all the comparisons (in all the tables), even if they are not
significant. The reason is that marginal differences or differences with a significance from 5% to 10% could be helpful to other researchers (i.e. The mean difference of table 3, after one year, in region one, is significant at 7%).
In table 2, remove the symbol (*) and meaning as is not used in the table data.
Reply: In regards to Table 1, we have supplemented the p-value data as you advise for readers' holistic understanding of the significance of data. We have also revised the legend in Table 2 for clarity.
Line 148: Reshape this sentence: "In the cohort of this extension study, ZA group retained higher mean BMD values in all Gruen zones at 5 years statistical significance."
Line 216: This study, although well-conducted, does not have the power to detect superiority. Data are not showing the confidence intervals to make readers hypothesize this. Please, consider reformulating this sentence.
Reply: Line 148 was reshaped for fluency (revised 158,159) and line 216 was revised to exclude the hypothetical statement in regards to power, with emphasis on the significance of bone turnover marker value over BMD values. (Revised line 223)
Line 269. A GEE was conducted at each time to test differences in all parameters.
Reply: Line 269 We have reworded the analysis using GEE and clarified with supplementation on the statistical methodology. (Revised 290-291)
Please, indicate if the regression model was univariate or multivariate. Whatever the case, specify the dependent and independent(s) variables(s).
Indicate the parameters used in the model in terms of link (identity, log, etc.), family (Gaussian, binomial, etc.) and type of correlation (unstructured, independent, exchangeable). Does the model fit? Please indicate the model adjustment information. If data were analyzed using univariate models, each table must show the model-adjustment information. Thank you!
Reply: Thank you for the detailed and stringent review and most valuable comments. Data within the study were analyzed with generalized estimating equation for each time-point when applicable as mentioned in the manuscript. Univariable analysis of the longitudinal data was performed with regression model for the study cohort in each regards including functional outcome (UCLA and HHS), radiographic outcome (BMD and change ratios) and laboratory values (serum and bone turnover markers).
No adjustment on the demography of patients were performed. We agree with you that the consideration of distribution and adjustment of data should be of concern with a small cohort such as our study. We are confident in the findings of our analysis, while being aware of standard errors associated with statistical analysis for a specific cohort. In hopes to emphasize the continual effect of Zoledronic acid on total hip arthroplasty, we did not pursuit to evaluate correlation between each time-point, but rather, observe the effect of Zoledronic acid, or lack of, on each variable in a progressive pattern. Our study has shown equitable statistical results and favorable clinical outcome. We appreciate greatly for your expertise and valued suggestions, and will take precaution and insight in further clinical analysis

Reviewer 3 Report
The study is well done. The subject is on high interest for surgicals teams.
Author Response
Reviewer#3
The study is well done. The subject is on high interest for surgical teams.
Reply: We thank the esteemed reviewer for your encouraging support and for the consideration in publication on Pharmaceuticals.

Reviewer 4 Report
The manuscript entitled "Zoledronic Acid ameiorates the bone turnover acrtivity an periprosthetic bone preservation in cement-less total hip arthroplasty" is well designe and well-written. If offers valuable information for the clinical practice future research. I recommend for publication, with minor revision.
1) Please add in the Discussion sector other bisphosphonates studied in this matter (ex. Alendronate, risedronate, and so on). Please discuss the advantages of zolendronic acid over these and compare your results.
2) Line 88: "No residual adverse effects were noted...".
Author Response
Reviewer#4
The manuscript entitled "Zoledronic Acid ameliorates the bone turnover activity an periprosthetic bone preservation in cement-less total hip arthroplasty" is well design and well-written. If offers valuable information for the clinical practice future research. I recommend for publication, with minor revision.
1) Please add in the Discussion sector other bisphosphonates studied in this matter (ex. Alendronate, risedronate, and so on). Please discuss the advantages of zolendronic acid over these and compare your results.
2) Line 88: "No residual adverse effects were noted...".
Reply: We thank the reviewer for your thorough and encouraging review, as well as the opportunity to clarify our manuscript.
1) In regards to the various bisphosphonates used clinically (Line 184), we have supplemented the revised manuscript discussing the favorable advantage of using zoledronate over alendronate for administration route and strategy, as well as the trend in the treatment of osteoporosis in orthopedic joint replacement. (Revisions 189-195).
2) Line 88: In this new version, we have revised to indicate the details. Thank you for your thorough review.

Round 2
Reviewer 2 Report
Dear Authors,
Thank you for the modifications to the manuscript.
Reviewer 4 Report
The revision has been made accordingly. I recommend for publication in the current form.